

# Clasnip: a web-based intraspecies classifier and multi-locus sequence typing for pathogenic microorganisms using fragmented sequences

Jiacheng Chuan[1,2], Huimin Xu[1], Desmond L. Hammill[1],
Lawrence Hale[2], Wen Chen[3,4] and Xiang Li[1]

[1] Charlottetown Laboratory, Canadian Food Inspection Agency, Charlottetown, Prince Edward Island, Canada
[2] Department of Biology, University of Prince Edward Island, Charlottetown, Prince Edward Island, Canada
[3] Department of Biology, University of Ottawa, Ottawa, Ontario, Canada
[4] Ottawa Research and Development Centre, Agriculture and Agri-Food Canada, Ottawa, Ontario, Canada

Corresponding author
Xiang Li, sean.li@inspection.gc.ca

## ABSTRACT

Bioinformatic approaches for the identification of microorganisms have evolved rapidly, but existing methods are time-consuming, complicated or expensive for massive screening of pathogens and their non-pathogenic relatives. Also, bioinformatic classifiers usually lack automatically generated performance statistics for specific databases. To address this problem, we developed Clasnip (www.clasnip.com), an easy-to-use web-based platform for the classification and similarity evaluation of closely related microorganisms at interspecies and intraspecies levels. Clasnip mainly consists of two modules: database building and sample classification. In database building, labeled nucleotide sequences are mapped to a reference sequence, and then single nucleotide polymorphisms (SNPs) statistics are generated. A probability model of SNPs and classification groups is built using Hidden Markov Models and solved using the maximum likelihood method. Database performance is estimated using three replicates of two-fold cross-validation. Sensitivity (recall), specificity (selectivity), precision, accuracy and other metrics are computed for all samples, training sets, and test sets. In sample classification, Clasnip accepts inputs of genes, short fragments, contigs and even whole genomes. It can report classification probability and a multi-locus sequence typing table for SNPs. The classification performance was tested using short sequences of 16S, 16–23S and 50S rRNA regions for 12 haplotypes of *Candidatus Liberibacter* solanacearum (CLso), a regulated plant pathogen associated with severe disease in economically important Apiaceous and Solanaceous crops. The program was able to classify CLso samples with even only 1–2 SNPs available, and achieved 97.2%, 98.8% and 100.0% accuracy based on 16S, 16–23S, and 50S rRNA sequences, respectively. In comparison with all existing 12 haplotypes, we proposed that to be classified as a new haplotype, given samples have at least 2 SNPs in the combined region of 16S rRNA (OA2/Lsc2) and 16–23S IGS (Lp Frag 4–1611F/Lp Frag 4–480R) regions, and 2 SNPs in the 50S rplJ/rplL (CL514F/CL514R) regions. Besides, we have included the databases for differentiating *Dickeya* spp., *Pectobacterium* spp. and *Clavibacter* spp. In addition to bacteria, we also tested

Clasnip performance on potato virus Y (PVY). 251 PVY genomes were 100% correctly classified into seven groups (PVY$^C$, PVY$^N$, PVY$^O$, PVY$^{NTN}$, PVY$^{N:O}$, Poha, and Chile3). In conclusion, Clasnip is a statistically sound and user-friendly bioinformatic application for microorganism classification at the intraspecies level. Clasnip service is freely available at www.clasnip.com.

## INTRODUCTION

Bioinformatic approaches for the identification of microorganisms have evolved rapidly, but existing methods are time-consuming, complicated, or expensive for massive screening of pathogens and their non-pathogenic relatives at interspecies or intraspecies levels. Average Nucleotide Identity (ANI) is widely used for the genome-wide comparison between closely related organisms (*Goris et al., 2007*), but obtaining the draft genome of unculturable bacteria usually requires deep sequencing along with hosts or environmental samples. Shotgun-based metagenomic classifiers can assist in the identification of closely-related organisms, but it also requires adequate genome coverage and a suitable algorithm for highly similar sequence identification (*Anyansi et al., 2020*). The cost of deep sequencing is still high, so the two methods mentioned are not ideal for massive and rapid screening and quarantine purposes. Amplicon-based metagenomic sequencing is possible for the classification at the intraspecies level, but its accuracy relies heavily on the choice of primer set (*Mancabelli et al., 2020*). A fast, sound and reliable bioinformatic classifier can facilitate the detection of pathogens, and it is important for disease containment and cost reductions for quick reaction in case of an outbreak or invasion of an exotic plant disease.

To meet the demand, we developed Clasnip (www.clasnip.com), a web-based platform for the classification and identification of closely related microorganisms on the basis of single nucleotide polymorphisms (SNPs) at intraspecies levels. Clasnip was programmed to classify selected DNA sequences of closely-related microorganisms, and generate an SNP table with a preprocessed database (Fig. 1). Users can build custom databases and evaluate classification performance using fragmented sequences or whole genome sequences (Fig. 1) other than the groups of pathogens listed in current databases. In this study, Clasnip was evaluated at the intraspecies level using datasets of *Candidatus* Liberibacter solanacearum (CLso) and potato virus Y (PVY).

CLso is an unculturable, phloem-limited bacterial pathogen. CLso is a regulated plant pathogen in European and Asian countries, and related to severe diseases in economically important Apiaceous and Solanaceous crops, such as potato, tomato, carrot, pepper, eggplant, tamarillo, tobacco, celery, and leek (*Sumner-Kalkun et al., 2020*). To date, 12 haplotypes of CLso have been identified around the world, including A (*Wen et al., 2009*), B (*Wen et al., 2009*), C (*Munyaneza et al., 2010*), D (*Nelson et al., 2013*), E (*Teresani et al., 2014*), F (*Swisher Grimm & Garczynski, 2019*), G (*Mauck et al., 2019*), H (*Haapalainen*

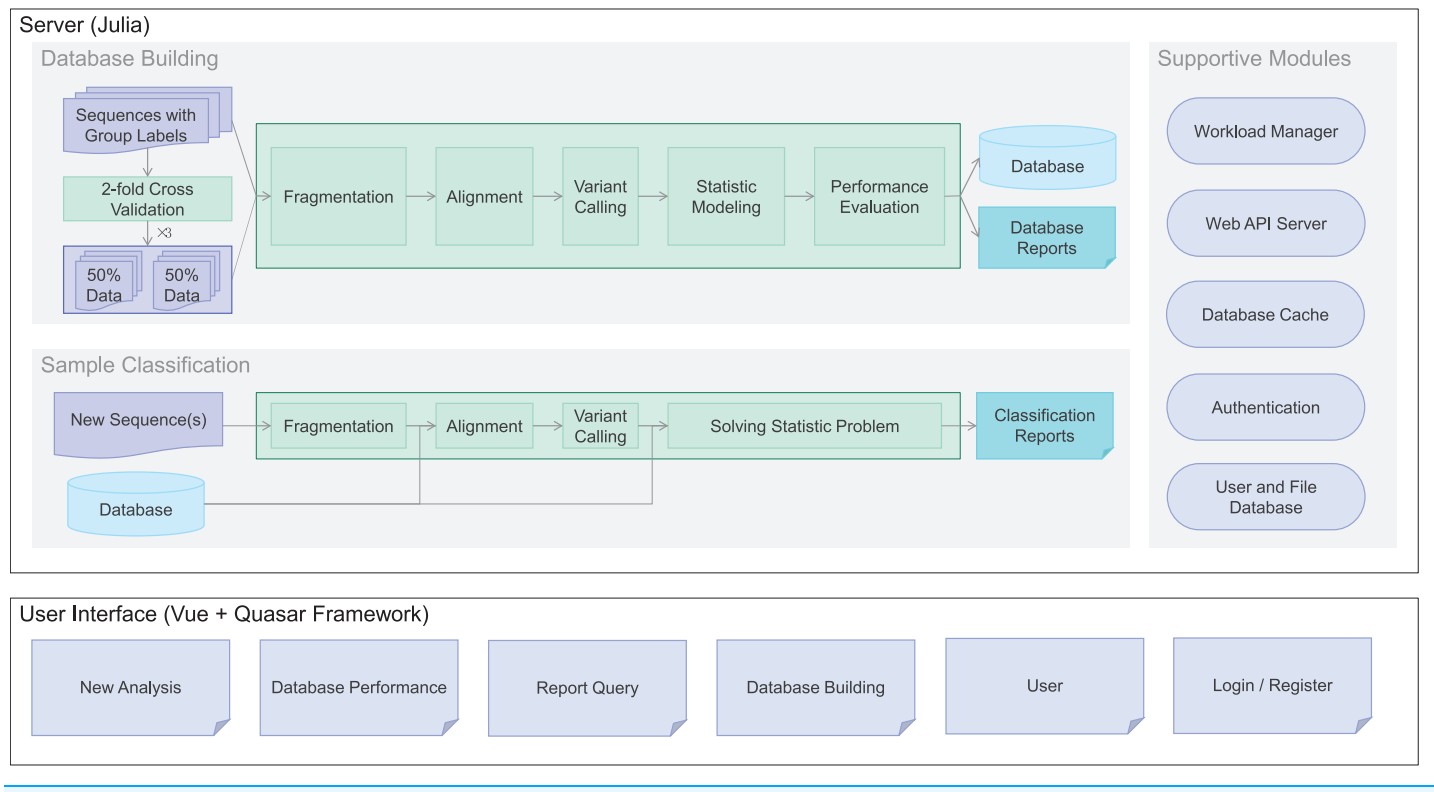

**Figure 1 Clasnip framework and analysis workflow.**

*et al., 2020*), H(Con) (*Contreras-Rendón et al., 2020*), U (*Haapalainen et al., 2018*), Cras1 (*Sumner-Kalkun et al., 2020*), and Cras2 (*Sumner-Kalkun et al., 2020*).

Potato virus Y (PVY) is a single-strand RNA virus. It is one of the most important regulated viruses causing disease of mainly Solanaceous plants globally (*Green et al., 2020*). The necrotic groups PVY$^N$ and PVY$^{NTN}$ cause veinal necrosis in tobacco, and PVY$^{NTN}$ also causes tuber necrosis in susceptible potato varieties (*Kehoe & Jones, 2016*). However, PVY$^C$ rarely infects potatoes and is inconsequential in the US potato crop (*Gray & Power, 2018*). Detection and diagnosis of PVY strains are difficult, and no single method can separate all strains into their respective types to date (*Potato Virus Y Strains, 2020*).

Thus, CLso and PVY are ideal species to test Clasnip for its performance at the intraspecies level because of their importance and complicated typing systems.

## MATERIALS AND METHODS

### Algorithms

The Clasnip performs sample classification based on a specialized database (Fig. 1). The Clasnip database is constructed with input samples with group labels, and one of the samples is marked as a reference. An input sample is a nucleotide sequence, or a collection of nucleotide sequences in the FASTA format. A sample can have sequences of a single gene, multiple genes, contigs, or even a genome. Users do not need to create any alignment file before submitting.

*Sample processing* is the first step of database building. This step aims to extract variants, including single nucleotide polymorphisms (SNPs), insertions, and deletions for each sample (Fig. 1). Long sequences in each sample are fragmented into 120-bp short sequences in the background, and two adjacent fragments have a 110-bp overlap. Then, all short fragments are mapped to the assigned reference with Bowtie2 v2.3.5.1, and argument '-k 10' is added to report alignments to up to ten loci (*Langmead et al., 2019*). Freebayes v1.3.2 was used for native polymorphism discovery (*Garrison & Marth, 2012*). The variant calling files generated in the VCF format are used to compute native variant frequencies for each group at each locus.

*Statistic modeling* is then used to guess classification groups based on variant frequencies at each location of each sample (Fig. 1). We assume the variant frequencies of each location operate as a discrete Markov chain. The classification groups are hidden from observations of variants, so the problem fits a Hidden Markov Model (HMM) and can be solved using the maximum likelihood method.

The variant frequency for each sample at a given location is computed based on variant depths. Given $D_{vk}$ as the depth of variant $v$ in sample $i$, $P_{vk}$ as the probability of variant $v$ in sample $i$ (ranging from 0 to 1):

$P_{vi} = D_{vi} / \sum_v D_{vi}$

Then, variant frequencies for a given group $g$ at a given locus (ranging from 0 to 1) are computed in the formula:

$$P_{vg} = \sum_i P_{vi} / \sum_v \sum_i P_{vi}$$

After the computation of the variant frequencies for all groups at all loci, if only one group is present at a locus, the frequencies at the locus will be removed to reduce database size. Then, the MLST of classification groups is written into the database.

*Sample classification.* Each input sample is classified using the new database. For each sample, only the common loci of the database and sample variants are used for group identity computation. For a given locus $l$, define $N_g$ as the maximum score of group $g$ that a sample can get:

$$N_g = \sum_l \max_v (P_{vlg})$$

This means, for a given locus and group, if only one variant exists, the maximum score is 1; if multiple variants exist, the maximum score is the maximum of variant probabilities related to variant depths, which is a punishment for non-unique variants.

Similarly, define $n_{sg}$ as the actual score of group $g$ that query sample $s$ gets:

$$n_{sg} = \sum_l \sum_v P_{vlg} \cdot W_{slv}$$

where $W_{slv}$ is the weight of variant $v$ at locus $l$ for query sample $s$, ranging from 0 to 1:

$$W_{slv} = D_{slv} / \sum_v D_{slv}$$

In this way, for a given locus $l$ and group $g$, if the query sample $s$ has a variant $v$ that the database does not, $P_{vlg} = 0$; if the database has a variant $v$, but the query sample does not, $W_{vls} = 0$. Thus, both cases do not contribute to $n_{sg}$.

Therefore, the sequence identity $I_{sg}$ for sample $s$ and group $g$ is defined as

$$I_{sg} = n_{sg}/N_g$$

The query sample is classified to the group with the highest identity.

After computing all group identities for input samples, Clasnip reports the 5%, 25%, and 50% quantiles of the group identities, and estimates the smoothed identity distribution for each group using the AverageShiftedHistograms package, an implementation of the Kernel Density Estimation over a fine-partition histogram in Julia language (*Joshday, 2021*). The identity distributions are stored for the estimation of the cumulated density (CDF) $F_g(I_{sg})$.

The estimated probability $p_{sg}$ of sample $s$ classified to group $g$ is normalized using the sum of the CDF of all groups:

$$p_{sg} = F_g(I_{sg}) \Big/ \sum_g F_g(I_{sg})$$

***Performance evaluation.*** Clasnip evaluates the classification performance of the new database. For a group $g$, True Positive (TP) is defined as the number of samples in group $g$ correctly classified to this group, which means no other group has higher identity than this group. False Positive (FP) is defined as the number of samples in group $g$ failed to be classified to this group, which means at least one group has higher identity than this group. True Negative (TN) is defined as the number of samples not in group $g$ classified to other groups. False Negative (FN) is defined as the number of samples not in group $g$ classified to group $g$. Then, performance statistics for each group are defined using the following formulas.

$$\text{TPR (sensitivity, recall, true positive rate)} = \frac{\text{TP}}{\text{TP} + \text{FN}}$$

$$\text{TNR (specificity, selectivity, true negative rate)} = \frac{\text{TN}}{\text{TN} + \text{FP}}$$

$$\text{PPV(precision, positive predictive value)} = \frac{\text{TP}}{\text{TP} + \text{FP}}$$

$$\text{NPV (negative predictive value)} = \frac{\text{TN}}{\text{TN} + \text{FN}}$$

$$\text{FNR (miss rate, false negative rate)} = 1 - \text{TPR}$$

$$\text{FPR (fall out, false positive rate)} = 1 - \text{TNR}$$

$$\text{FDR (false discovery rate)} = 1 - \text{PPV}$$

$$\text{FOR (false omission rate)} = 1 - \text{NPV}$$

$$\text{ACC (accuracy)} = \frac{(\text{TP} + \text{TN})}{(\text{TP} + \text{TN} + \text{FP} + \text{FN})}$$

$$F_1 = 2\frac{\text{PPV} \times \text{TPR}}{\text{PPV} + \text{TPR}}$$

Besides, the overall accuracy of the database is the ratio of correctly classified samples to all samples.

Wrongly classified samples and low-coverage (<5 SNPs) samples are reported on the database detail page in Clasnip. It allows the database maintainer to check the data quality and classification performance of the reference region.

The previous database validation uses all samples as training and test sets. To avoid overfitting, we also perform three replicates of stratified two-fold cross-validation using the methods above for single-gene databases. Filtrations are applied for cross-validation: (1) low-coverage samples with less than five SNPs are removed; (2) groups with only one sample are ignored. After filtration, samples within each group are randomly partitioned into two equal-sized subsets $d_0$ and $d_1$. Then, Clasnip trains on $d0$ and tests on $d1$, followed by training on $d1$ and testing on $d0$. The mean and standard deviation of all performance statistics of the training and test sets are computed, respectively.

In addition, a heatmap of group similarity based on sample identity is computed for whole-genome or multi-gene databases. For any sample labeled as group $g_0$, it has a set of identity values $I$ compared with each group $g$. Let denote $I_{\overline{g_0,g}}$ as the mean identity of all samples labeled as group $g_0$ compared to group $g$. Then, we can build a mean identity matrix with $g_0$ as the x-axis, and $g$ as the y-axis. Then, the pairwise Euclidean distance is computed, and clustering is performed. The average identity indicated in the heatmap is related to the samples' mapped regions to the reference.

For a single-gene database, Clasnip plots the heatmap of group similarity based on SNP differences. The heatmap is a two-dimensional matrix, with the x- and y-axis as groups. Two values are shown in each cell: (1) SNP score and (2) total SNP number. SNP score is defined as the sum of the variant probabilities $P$ of unique SNPs between two groups throughout the overlapped region. The SNP score can also be regarded as the weighted number of distinct SNPs of two groups, and the weight is the variant's frequency among the samples in one group. Total SNP number means the total number of SNPs among all groups in the overlapped region. The fraction of the SNP score and total SNP number is used for cell coloring and clustering of the heatmap.

## Real plant materials

A total of 20 tomato plants (two batches, J1–J10 and M1–M10) were planted for four weeks, and then each plant was treated with a collection of psyllids for 5–7 days. The psyllids are vectors of CLso haplotypes A and B. Four weeks after inoculation, the

Table 1 Sequence summary of the curated CLso haplotype database.

| Region | A | B | C | Cras1a* | Cras1b* | Cras2 | D | E | F | G | H | H-Con | U | Grand total |
|---|---|---|---|---|---|---|---|---|---|---|---|---|---|---|
| Genomic | 4 | 1 | 2 | | | | 1 | | | | | | | 8 |
| 16S | 5 | 8 | 6 | 13 | 3 | 3 | 9 | 5 | 1 | 3 | 1 | 2 | 1 | 60 |
| 16S, 16–23S | 1 | | | | | | | | | | | | | 1 |
| 16S, 16–23S, 5S | | | 3 | | | | | | | | | | | 3 |
| 16–23S | 1 | | 16 | 15 | 3 | 4 | 23 | 7 | | 3 | 1 | | 1 | 74 |
| 50S | 2 | 2 | 29 | 18 | 3 | 4 | 16 | 8 | 1 | 4 | 1 | | 5 | 93 |
| adk | | | 31 | | | | 1 | | | 4 | 1 | | 5 | 42 |
| atpA | | | 31 | | | | 1 | | | 4 | 1 | | 5 | 42 |
| fbpA | | | 31 | | | | 1 | | | 4 | 1 | | 5 | 42 |
| ftsZ | | | 30 | | | | 1 | | | 4 | 2 | | 5 | 42 |
| glyA | | | 31 | | | | 1 | | | 4 | 1 | | 5 | 42 |
| groEL | | | 31 | | | | 1 | | | 4 | 1 | | 5 | 42 |
| gyrB | | | 31 | | | | 1 | | | 4 | 2 | | 5 | 43 |
| omp | | | | 11 | 3 | 4 | | 1 | | | | | | 19 |
| Grand total | 13 | 11 | 272 | 57 | 12 | 15 | 56 | 20 | 3 | 38 | 12 | 2 | 42 | 553 |

Note:
* Cras1a and Cras1b are sub-categories of haplotype Cras1.

DNA of stems and leaves were extracted and purified using MagneSil KF, a genomic extraction kit, following the manufacturer's protocol. All tomato plants were tested with CLso PCR primers CLipoF/OI2c (*Liefting et al., 2009*; *Secor et al., 2009*) (16S rRNA), and the ten plants in the second batch were also tested with the 50S rplJ/rplL primers (CL514F/CL514R (*Munyaneza et al., 2009*)). PCR gel products of positive samples were extracted using QIAquick Gel Extraction Kit according to the manufacturer's protocol and sent out for Sanger sequencing at the Ottawa Hospital Research Institute. The raw sequences were trimmed using a 0.01 quality score in CLC Genomics Workbench 20.0.2. After trimming, clean sequences were used for haplotype identification using Clasnip.

## RESULTS

### Clasnip databases for CLso pathogen

A total of 553 CLso sequence records with clear haplotype metadata were fetched from the NCBI database to build Clasnip CLso databases (Table 1). The detailed accession numbers are listed in Table S1. This database comprises 12 haplotypes, except that the haplotype Cras1 is decomposed to Cras1a and Cras1b groups. Haplotypes A, B, C and D contain at least one whole-genome sequencing datasets (Table 1). All haplotypes have 16S rRNA sequences included (Table 1 and Fig. 2). Haplotypes F and H(Con) do not have records of the 16–23S intergenic space sequence, and haplotype H(Con) lacks the record of 50S rRNA sequence (Table 1 and Fig. 2). In addition, the *adk, atpA, gbpA, ftsZ, glyA, groEL, gyrB* genes do not have records in haplotypes Cras1, Cras2, E, F, and H(Con) (Table 1).

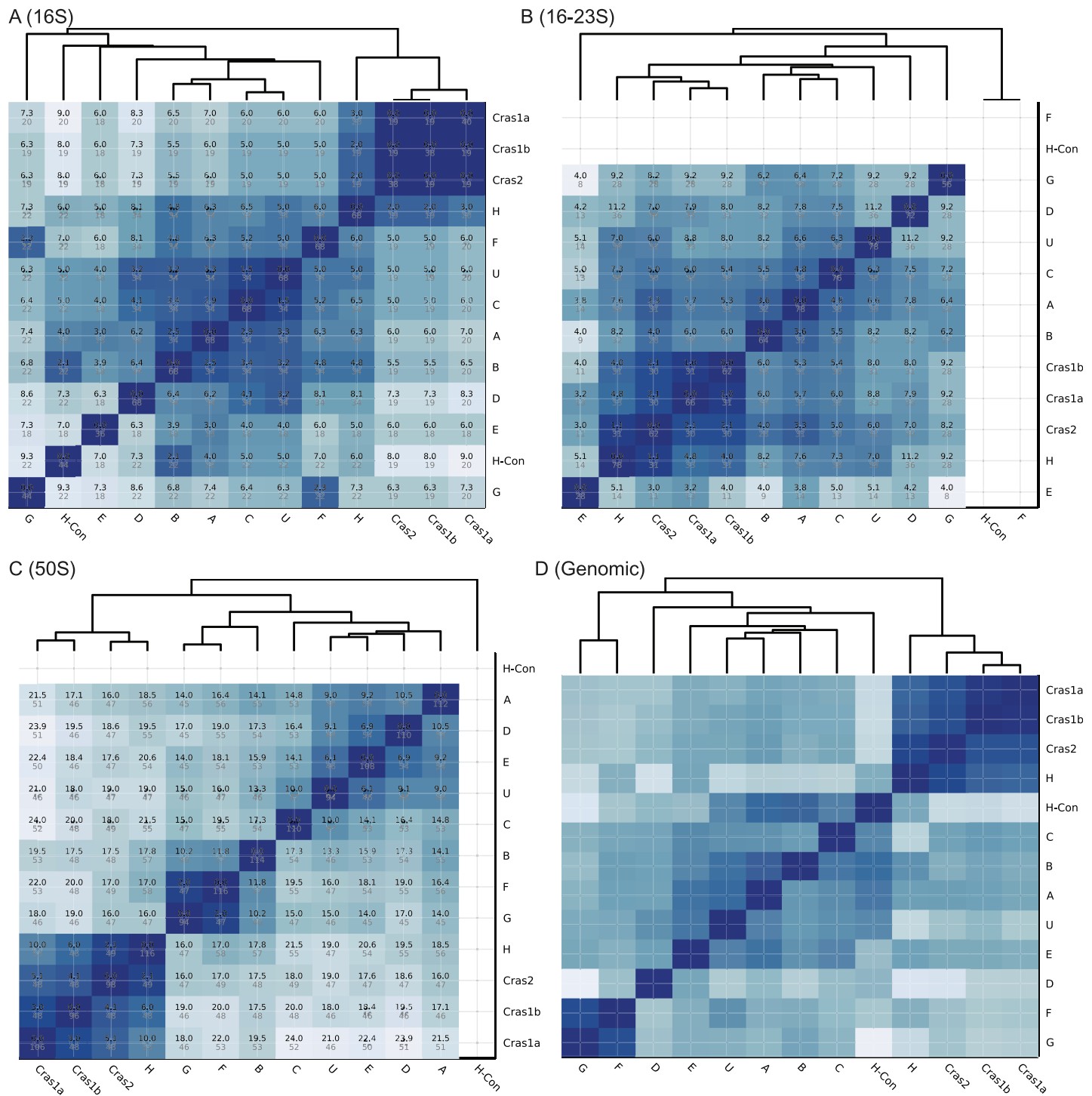

**Figure 2 Heatmaps of CLso haplotype similarity in different genomic regions.** Darker color means higher similarity. (A–C) The similarity of single gene database is based on SNP differences. A number in black is the SNP score, which is the sum of frequencies of unique SNPs in the overlapped region between two haplotypes. A number in grey is the total number of SNPs in the overlapped region among all haplotypes. (D) The similarity of the genomic database is based on sequence identity.

**Table 2 Clasnip classification performance of CLso rRNA gene regions.**

| Group | 16S | | | 16-23S IGS | | | 50S rplJ/rplL | | |
| --- | --- | --- | --- | --- | --- | --- | --- | --- | --- |
| | # Sample | Identity (Q5) (%) | Accuracy (%) | # Sample | Identity (Q5) (%) | Accuracy (%) | # Sample | Identity (Q5) (%) | Accuracy (%) |
| A | 10 | 93.4 | 100.0 | 6 | 90.8 | 100.0 | 6 | 97.6 | 100.0 |
| B | 9 | 92.4 | 88.9 | 1 | 97.5 | 100.0 | 3 | 97.3 | 100.0 |
| C | 11 | 93.7 | 90.9 | 21 | 90.0 | 100.0 | 31 | 100.0 | 100.0 |
| Cras1a | 13 | 100.0 | 100.0 | 15 | 95.8 | 93.3 | 18 | 99.8 | 100.0 |
| Cras1b | 3 | 100.0 | 100.0 | 3 | 100.0 | 100.0 | 3 | 99.8 | 100.0 |
| Cras2 | 3 | 100.0 | 100.0 | 4 | 98.6 | 100.0 | 4 | 99.1 | 100.0 |
| D | 10 | 98.2 | 100.0 | 24 | 87.4 | 100.0 | 17 | 96.4 | 100.0 |
| E | 5 | 100.0 | 100.0 | 7 | 92.0 | 100.0 | 8 | 98.6 | 100.0 |
| F | 1 | 100.0 | 100.0 | – | – | – | 1 | 100.0 | 100.0 |
| G | 3 | 98.3 | 100.0 | 3 | 98.9 | 100.0 | 4 | 100.0 | 100.0 |
| H | 1 | 100.0 | 100.0 | 1 | 100.0 | 100.0 | 1 | 100.0 | 100.0 |
| H-Con | 2 | 100.0 | 100.0 | – | – | – | – | – | – |
| U | 1 | 100.0 | 100.0 | 1 | 100.0 | 100.0 | 5 | 99.8 | 100.0 |
| Total | 72 | – | 97.2 | 86 | – | 98.8 | 101 | – | 100.0 |

**Note:**
# Sample is the number of samples with more than 5 SNPs covered in the reference region. Identity (Q5) means the 5% quantile of estimated identity distribution. If a new sample's identity is greater than the identity (Q5) of a group, the new sample is classified into the group. Accuracy is the ratio of correctly classified samples to all samples with more than 5 SNPs covered in the reference region. "Correctly classified" is defined as the identity of the sample's group is the highest among other groups.

We built four databases using different reference sequences: (1) 16S rRNA database (NCBI accession number MH259699) (2) 16–23S rRNA database (NCBI accession number JX624236) (3) 50S rRNA database (NCBI accession number MH259700) and (4) whole genome database (NCBI assembly number ASM18366v1).

The heatmaps of the four databases are shown in Fig. 2. Based on the 16S rRNA sequences, haplotypes Cras1 and Cras2 are identical with no sequence differences, and haplotypes C and U have only 1.5 SNP score (Fig. 2A). In the 16–23S region, the closest groups are Cras1a and Cras1b with only 1.1 SNP score (Fig. 2B). In the 50S region, each group pair has more distinct SNPs than the 16S and 16–23S regions, and the closest groups are Cras2 and H (2.1 SNP score) (Fig. 2). The heatmap of the genomic database does not show SNP scores because the genomic coverages of haplotypes are divergent (Table 1). Regions in a whole genome can be variable or conserved, which makes similarity based on SNP count not suitable for genome-wide databases.

The classification performance of the CLso genomic database is shown in Table 2. In the 259 samples with enough coverage (>5 SNPs) in the 16S, 16–23S or 50S rplJ/rplL regions, Clasnip successfully classified 256 samples based on the 16S rRNA, 16–23S intergenic spacer region and 50S rplJ/rplL gene sequences, resulting in 97.2%, 98.8% and 100% accuracy, respectively (Table 2). Only two out of 72 samples in the 16S region are misclassified. One misclassification is due to the SNP variations of the different copies of the same genome in the sample ASM18366v1 of haplotype B, which has three copies of the 16S rRNA gene with SNPs variations. Another one is the 16S sample MG701017 of haplotype C. It has 98.1% identity, 31.0/31.6 SNP score, and 82.1% CDF with haplotype C,

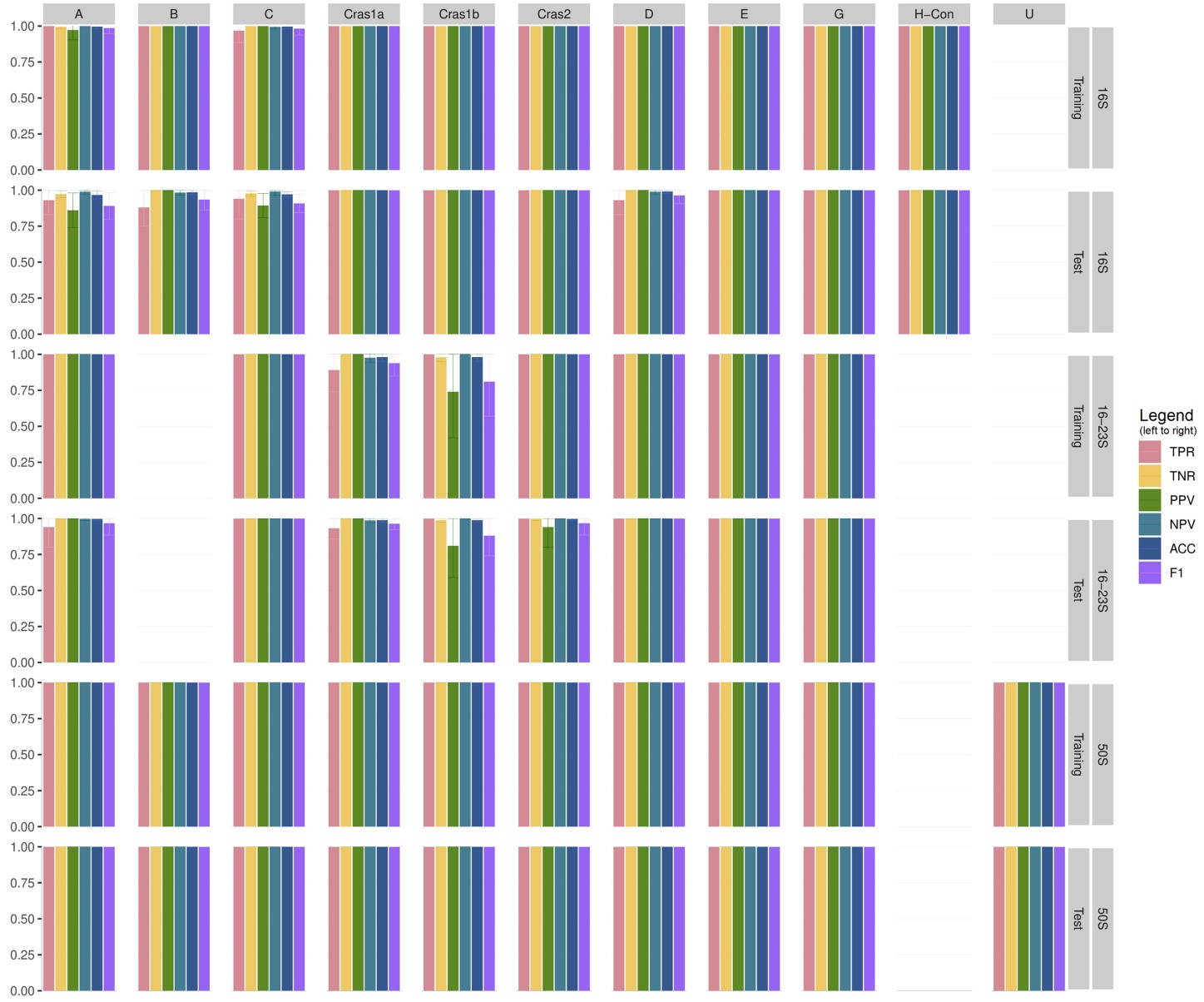

**Figure 3 Cross-validation performance of CLso 16S, 16–23S and 50S rRNA databases.** TPR = Sensitivity, Recall, Hit Rate, True Positive Rate. TNR = Specificity, Selectivity, True Negative Rate. PPV = Precision, Positive Predictive Value. NPV = Negative Predictive Value. ACC = Accuracy. F1 = the harmonic mean of precision and sensitivity.

but it also has 100% identity, 34.0/34.0 SNP score, and 100.0% CDF with haplotype U. Clasnip reports the sample with both C and U groups with the probability of 37.2% and 45.3%, respectively. In addition, one out of 86 samples in the 16–23S region is misclassified (Table 2). The sample only contains half of the normal 16–23S region with many sequencing errors.

Table S2 shows the performance statistics of the CLso 16S, 16–23S, 50S rRNA and genomic databases. Figure 3 and Table S3 show the cross-validation statistics of the databases. The performance of the 50S rRNA database in both training and test sets

**Table 3 Performance benchmark of Clasnip and BLCA using CLso 16S rRNA sequences.**

| Program | TPR (%) | TNR (%) | PPV (%) | NPV (%) | ACC (%) | F1 (%) |
|---------|---------|---------|---------|---------|---------|--------|
| Clasnip | 99.1 | 99.9 | 96.2 | 99.9 | 99.8 | 97.0 |
| BLCA | 75.5 | 98.6 | 75.7 | 98.3 | 97.2 | 74.9 |

Note:
The values of statistics are shown as the mean of all CLso haplotypes. TPR = Sensitivity, Recall, Hit Rate, True Positive Rate. TNR = Specificity, Selectivity, True Negative Rate. PPV = Precision, Positive Predictive Value. NPV = Negative Predictive Value. ACC = Accuracy. F1 = the harmonic mean of precision and sensitivity.

achieves 100% sensitivity and specificity in all groups (Fig. 3 and Table S3). The 16S and 16–23S rRNA regions of some haplotypes are similar, with less than 2 SNPs (Fig. 2), and the general performance is not as good as the 50S rRNA database, especially for groups with less than five samples (Fig. 3, Tables S2 and S3).

## Comparison to related work

Clasnip classifies gene or genomic sequences in the FASTA format using HMM and Maximum Likelihood Estimation. The classification pipeline used in Clasnip can be benchmarked with other sequence classification tools. BLCA is a taxonomy classification program based on a Bayesian-based lowest common ancestor (LCA) method and has favorable species-level accuracy for 16S rRNA classification (*Gao et al., 2017*). The custom BLCA database was built using 64 16S rRNA sequences, and whole genome sequences were not included because whole genome classification was out of the scope of BLCA (Table S1). The performance of the 64 samples was evaluated using the same method as Clasnip (Tables S4 and S5). The mean performance of all haplotypes is shown in Table 3. The sensitivity and specificity of Clasnip are 99.1% and 99.9%, respectively (Table 3). In comparison, BLCA has 75.5% sensitivity and 98.6% specificity (Table 3). Therefore, the algorithm behind Clasnip is favorable in the classification at the haplotype level.

## Real sample classification

While CLso haplotypes A and B are both lethal to potato plants, the symptoms of tomato plants infected by CLso haplotypes A and B are significantly different. Tomato plants infected by CLso haplotype B declined fast and died after being infected for 4–8 weeks, while plants infected by CLso haplotype A can maintain viability without apparent symptoms for multiple years (*Li et al., 2013*). Therefore, it is practically useful to differentiate haplotypes A and B for proper management of the disease appearance in tomato plants, which could become the primary inoculum for the vector potato/tomato psyllid to transmit and spread potato zebra chip disease.

We tested the CLso Clasnip database for new sample classification using 20 tomato plants infected by CLso haplotype A or B (Table 4 and Table S6). All tomato plants were tested with CLso PCR primers CLipoF/OI2c (*Liefting et al., 2009*; *Secor et al., 2009*) (16S rRNA), and ten plants were also tested with the 50S rplJ/rplL primers (CL514F/CL514R (*Munyaneza et al., 2009*)) (Table 4). The sequences are available at Clasnip in Table S6. Samples J1, J4, J7, and J10 tested positive in PCR, but the concentrations of PCR products were too low to be sequenced. Sample M4 tested negative in PCR. In the rest of the

**Table 4 The classification result of real tomato samples against the CLso genomic database.**

| Sample | Real haplotype | Region | Group Rank | Group | Identity (%) | Matched SNP Score | Covered SNP Score | CDF (%) | Probability (%) |
|---|---|---|---|---|---|---|---|---|---|
| J2 | A | 16S | 1 | A | 100.0 | 34.0 | 34.0 | 100 | 62 |
| | | | 2 | B | 97.8 | 32.9 | 33.7 | 60 | 37 |
| J3 | A | 16S | 1 | A | 100.0 | 18.0 | 18.0 | 100 | 69 |
| | | | 2 | B | 95.8 | 16.9 | 17.7 | 30 | 21 |
| | | | 3 | H-Con | 94.4 | 17.0 | 18.0 | 15 | 11 |
| J5 | A | 16S | 1 | A | 100.0 | 34.0 | 34.0 | 100 | 62 |
| | | | 2 | B | 97.8 | 32.9 | 33.7 | 60 | 37 |
| J6 | A | 16S | 1 | A | 100.0 | 33.0 | 33.0 | 100 | 84 |
| | | | 2 | B | 94.6 | 30.9 | 32.7 | 18 | 15 |
| J8 | A | 16S | 1 | A | 100.0 | 32.0 | 32.0 | 100 | 63 |
| | | | 2 | B | 97.6 | 30.9 | 31.7 | 58 | 36 |
| J9 | A | 16S | 1 | A | 100.0 | 34.0 | 34.0 | 100 | 83 |
| | | | 2 | B | 94.8 | 31.9 | 33.7 | 19 | 16 |
| M1 | B | 16S + 50S | 1 | A | 93.8 | 75.0 | 80.0 | 15 | 60 |
| | | | 2 | B | 93.7 | 74.4 | 79.4 | 11 | 40 |
| M2 | B | 16S + 50S | 1 | B | 100.0 | 81.4 | 81.4 | 100 | 100 |
| M3 | B | 16S + 50S | 1 | B | 100.0 | 81.1 | 81.1 | 100 | 100 |
| M5 | B | 16S + 50S | 1 | B | 96.9 | 74.1 | 76.4 | 47 | 94 |
| M6 | B | 16S + 50S | 1 | B | 97.5 | 77.4 | 79.4 | 55 | 98 |
| M7 | B | 16S + 50S | 1 | B | 100.0 | 78.4 | 78.4 | 100 | 100 |
| M8 | B | 16S + 50S | 1 | B | 100.0 | 79.4 | 79.4 | 100 | 100 |
| M9 | A | 16S + 50S | 1 | A | 98.2 | 76.6 | 78.0 | 72 | 100 |
| M10 | B | 16S + 50S | 1 | B | 98.8 | 81.4 | 82.4 | 77 | 100 |

**Note:**

Samples J1, J4, J7 and J10 were tested positive in PCR, but the concentrations of PCR products were too low to be sequenced. Sample M4 was tested negative in PCR. Identity is the ratio of matched SNP score to covered SNP score. CDF is the cumulated density where the sample falls in the estimated distribution of the database. Probability is the classification likelihood. Only rows with probability greater than 10% are shown.

15 samples, 14 samples, except for sample M1, were clearly classified to the right haplotype in terms of identity (Table 4). The identity between M1 and haplotype A is 93.8%, only 0.1% higher than haplotype B (Table 4). The identity values are lower than other samples, which could be caused by sequencing errors or the existence of multiple infections by both CLso haplotype A and B.

## Clasnip database for potato virus Y

Though PVY has different complex typing systems, we chose to build a PVY database for practical quarantine needs, using 251 whole viral genomes belonging to phylotypes of PVY$^N$, PVY$^O$, PVY$^C$, PVY$^{NTN}$, PVY$^{N:O}$, Poha strains (a close strain group within PVY$^C$), and Chile3 strain (a strain that is distinct from all other PVY strains) (*Green et al., 2020*; *Kehoe & Jones, 2016*; *Gibbs et al., 2017*). The accession numbers of PVY whole genomes are listed in Table S7. The reference strain of the database is HQ912865.1.

Figure 4 shows the genomic similarity heatmap of the PVY database. A closer relation was found in PVY$^{NTN}$, PVY$^{N:O}$ and PVY$^O$ (Fig. 4). Poha was close to PVY$^C$, according to

 

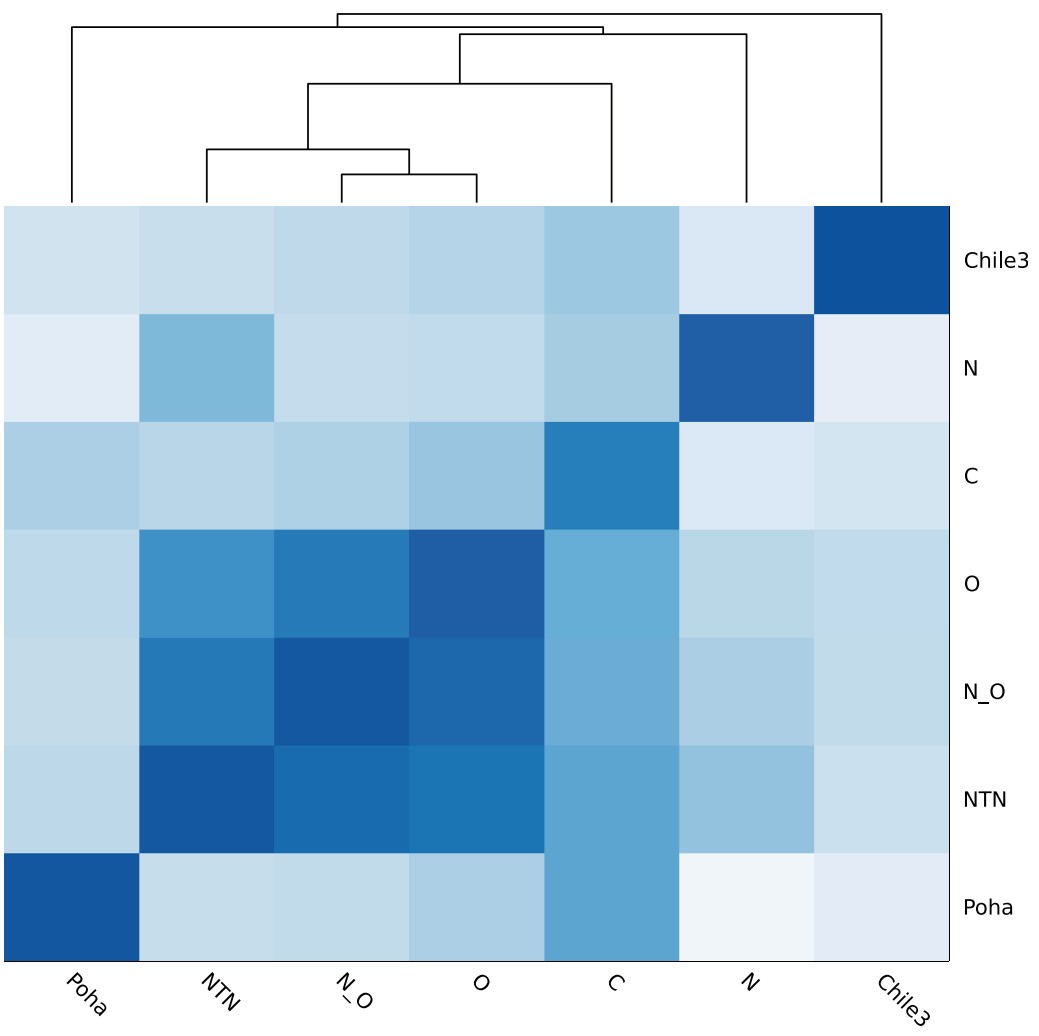

**Figure 4 Heatmap of genomic similarity of PVY phylogroups.** Darker color means higher similarity.

the dark color between Poha and PVY$^C$ (Fig. 4). All PVY genomes were correctly classified (Table 5 and Table S2). Even in the cross-validation test, the PVY samples were 100% correctly classified (Table S3).

## DISCUSSIONS

Clasnip provides a user-friendly interface and robust implementation of the Hidden Markov Model for bioinformaticians and non-bioinformaticians to classify closely related pathogens at the intraspecies level. The input of Clasnip is not limited to short sequences from sanger sequencing, but also the assembled long sequences from next-generation sequencing. Users do not need to align sequences prior to Clasnip. Clasnip can not only provide sample classification at the intraspecies level, but also a tool to measure the similarity of genes originating from plant pathogens with populational variations. The single-gene database of Clasnip supports SNP-based differentiation, and the
**Table 5 Clasnip classification performance of PVY database.**

| Group | # Sample | Identity (Q5) (%) | Accuracy (%) |
|---|---|---|---|
| C | 16 | 85.8 | 100.0 |
| Chile3 | 1 | 99.9 | 100.0 |
| N | 31 | 92.4 | 100.0 |
| NTN | 73 | 96.0 | 100.0 |
| N:O | 37 | 96.8 | 100.0 |
| O | 87 | 95.0 | 100.0 |
| Poha | 6 | 97.4 | 100.0 |
| Total | 251 | – | 100.0 |

Note:
# Sample is the number of samples with more than five SNPs covered in the reference region. Identity (Q5) means the 5% quantile of estimated identity distribution. If a new sample's identity is greater than the identity (Q5) of a group, the new sample is classified into the group. Accuracy is the ratio of correctly classified samples to all samples with more than 5 SNPs covered in the reference region. "Correctly classified" is defined as the identity of the sample's group is the highest among other groups.

performance of classification at the intraspecies level for CLso haplotypes and PVY phylogroups is readily achieved.

Currently, haplotyping of CLso is mainly based on the manual check using multi-locus sequence typing (MLST), and there is no standardized software for fast and comprehensive haplotyping for CLso, and no comprehensive CLso database containing curated haplotype labels. Furthermore, the current MLST analysis of CLso has limitations: (i) The performance of MLST classification is unknown. (ii) Only one or two sequences were selected for each haplotype and gene (*Sumner-Kalkun et al., 2020*; *Nelson et al., 2013*; *Swisher Grimm & Garczynski, 2019*; *Contreras-Rendón et al., 2020*; *Haapalainen et al., 2018*), which might neglect some SNPs within the same haplotype. (iii) Some haplotypes lack sequences of common rRNA and conserved regions, such as haplotype H (Con) lacking 16–23S IGS and 50S rplJ/rplL rRNA regions (*Contreras-Rendón et al., 2020*), and haplotype F lacking 16–23S IGS region (*Swisher Grimm & Garczynski, 2019*). (iv) For one gene in MLST tables published in different papers, differences exist in starting positions, reference sequences, and SNP notations, which hinder comparison and validation (*Swisher Grimm & Garczynski, 2019*; *Haapalainen et al., 2018*).

Clasnip solves the above problems and provides solid haplotype lineage and similarity analysis for the classification of CLso haplotypes.

Hence, based on the SNP profiles, we propose the following rule to identify new CLso haplotypes. A new haplotype should have at least 2 SNPs in the combined region of 16S rRNA (OA2/Lsc2 (*Liefting et al., 2009*; *Haapalainen et al., 2017*)) and 16–23S IGS (Lp Frag 4–1611F/Lp Frag 4–480R (*Hansen et al., 2008*)) regions, and 2 SNPs in the 50S rplJ/rplL (CL514F/CL514R (*Munyaneza et al., 2009*)) region, comparing to existing haplotypes. If a strain does not meet the requirement, it should be considered a variant of an existing haplotype.

In addition to CLso, Clasnip also has an excellent performance in the classification of potato virus Y phylogroups. The Clasnip website also provides classification services for *Dickeya* spp., *Pectobacterium* spp., and *Clavibacter* spp. (data not showing).

## CONCLUSIONS

Clasnip has been released to the public as an easy-to-use web-based application for the classification of closely related microorganisms at the intraspecies level, which is hosted at www.clasnip.com. The program is rigorously tested using public and private CLso sequences, as well as potato virus Y sequences, proving the performance of Clasnip. Moreover, we assessed the CLso haplotype similarity based on SNP analysis using Clasnip and proposed a general guide for the determination of new CLso haplotypes.

### Availability and requirements

– Project name: Clasnip.

– Project home page: http://www.clasnip.com.

– Project source code, sequence data and database files are available at the GitHub repository: https://github.com/cihga39871/clasnip_data (DOI 10.5281/zenodo.7294958).

– Operating system of host server: Linux.

– Programming language: Julia v1.8 (https://julialang.org/), Vue.js v2 (https://vuejs.org/).

– Other requirements: Quasar Framework v1.20.1 (https://quasar.dev/).

– Parallel implementation and workload management: Julia packages JobScheduler.jl v0.7.1 (https://github.com/cihga39871/JobSchedulers.jl), Pipelines.jl v0.8.5 (https://github.com/cihga39871/Pipelines.jl).

## ACKNOWLEDGEMENTS

The technical assistance of Jingbai Nie is greatly acknowledged. The advice on algorithm optimization, encouragement, and support of Drs. Christian Lacroix and Stevan Springer to JC are greatly appreciated.

### Funding

This study was funded by the Interdepartmental fundings of Living Laboratories Initiatives, Atlantic Project, and Genomics Research and Development Initiatives Project to Xiang Li. There was no additional external funding received for this study. The funders had no role in study design, data collection and analysis, decision to publish, or preparation of the manuscript.

### Grant Disclosures

The following grant information was disclosed by the authors:
Interdepartmental fundings of Living Laboratories Initiatives, Atlantic Project, and Genomics Research and Development Initiatives Project.

### Competing Interests

The authors declare that they have no competing interests.

## Author Contributions

- Jiacheng Chuan conceived and designed the experiments, performed the experiments, analyzed the data, prepared figures and/or tables, authored or reviewed drafts of the article, and approved the final draft.
- Huimin Xu analyzed the data, authored or reviewed drafts of the article, and approved the final draft.
- Desmond L. Hammill performed the experiments, analyzed the data, authored or reviewed drafts of the article, and approved the final draft.
- Lawrence Hale analyzed the data, authored or reviewed drafts of the article, and approved the final draft.
- Wen Chen analyzed the data, authored or reviewed drafts of the article, and approved the final draft.
- Xiang Li conceived and designed the experiments, analyzed the data, authored or reviewed drafts of the article, and approved the final draft.

## Data Availability

The data and code are available at GitHub: https://github.com/cihga39871/clasnip_data; Jiacheng Chuan. (2022). cihga39871/clasnip_data: v1.0.1 (v1.0.1). Zenodo. https://doi.org/10.5281/zenodo.7294958.

## Supplemental Information

Supplemental information for this article can be found online at http://dx.doi.org/10.7717/peerj.14490#supplemental-information.

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
