# Peer review of "Clasnip: a web-based intraspecies classifier and multi-locus sequence typing for pathogenic microorganisms using fragmented sequences"

_PeerJ, doi:10.7717/peerj.14490_

## Round 0.1 · original submission · Major Revisions

The two reviewers were generally positive about the work. However, each made a comment that I think needs to be addressed. Reviewer 1 pointed out the need to test the software on other datasets. And Reviewer 2 mentioned that Clasnip should be compared to at least one other competing software.

In addition to addressing these two specific comments, you should also address all other points raised.

Reviewer 1 ·

Basic reporting

The article is written well and is easy enough to follow. However, the value of Clasnip can be highlighted a bit better. Currently, there is mention of distinguishing pathogens from their non-pathogenic relatives or sub-species, especially for quarantine purposes. Maybe elaborate on specific organisms referred to here and why it would be important to be able to distinguish these from one another using clasnip. Most diagnostic laboratories dealing with quarantine organisms will have specific tests available for the organism of importance, and a positive result is sufficient to take action against a quarantine pathogen. Why would they then need to further validate the results with clasnip and more sequencing? I would rather think that clasnip will be a good tool to use for identifying populations.

Specific grammar suggestions:
Abtract:
Line 23: change closely related microorganisms to relatives
Line 23-24: Consider deleting 'for quarantine purposes'
Line 25: 'To address the' change to 'address this'
Line 26: Change to 'web-based'
Line 38: Change to '(Clso), a regulated plant pathogen associated with severe disease'
Line 39-40: Delete ' such as potato, tomato and carrot'
Line 42-43: Consider re-writing as: we propose that to be classified as a new haplotype, a given samples have at least 2 SNPs .......
Line 47: replace sub-species with haplotype. There are other factors to be considered when deciding whether or not you are dealing with a subspecies, additionally, from the experiments conducted you were only able to show that clasnip is able to differentiate haplotypes.

Introduction:
Line 53: rewrite as: non-athogenic relatives at species etc
Line 54: delete quarantine.
Line 59-61: Talks about quarantine, see comments above and consider adding specific examples of where clasnip will be used in quarantine facilities.
Line 63: You speak of detection of pathogens without giving examples of specific pathogens which have a few SNP differences when compared to a non-pathogenic relative. Again, will clasnip not be better suited to assist with population studies? Can clasnip maybe not be used for viral strain identification? e.g Citrus tristeza virus (CTV) has a number of strains and can present in a mixed infection, some being mild and others causing severe disease. Can clasnip not be able to help differentiate the different strains in a population?

Materials and methods:
Line 86: based on a
Line 95-96: This sentence is a bit confusing, does the window only show 120bp of a given sequence at a time? So do you have to re-open the window every 120bp for larger sequences?
Line 117: written

Discussion
Line 319: replace sub-species with haplotypes

Experimental design

The experiment was well though out, but maybe consider testing clasnip on different datasets as this will help showcase the ability and reproducibility of the software. Maybe consider doing tests on CTV databases as well as something like Xylella subspecies. This also help showcase that clasnip can be used for different pathogens.

Validity of the findings

No comment

Additional comments

No comment

Reviewer 2 ·

Basic reporting

Overall, the contents and topic of the paper are solid. The authors provide good background information and motivations for the development of their Clasnip tool, which makes the paper easy to follow. The statistics behind Clasnip are explained thoroughly and are well-established in bioinformatics (e.g. HMMs are used very often). There were just some grammar mistakes sprinkled throughout and other technicalities mentioned below.
The paper lacks a data availability statement with the tomato plant dataset used that can be accessed from outside the Clasnip website. A data availability statement would make it easier to identify where the data can be accessed, because searching for the S1 supplementary table with the accession numbers is not optimal. Maybe the statement can link to the table or to clasnip_data/data at master · cihga39871/clasnip_data (github.com). The Github page would benefit from more usage documentation regarding setup and usage. A description of the steps included within the pipelines would be very useful, especially as a visual aid. The website seems to work as described.

Experimental design

The experimental design seems well-defined and logical. A dataset with known haplotypes of CLso was used to evaluate Clasnip's classification of haplotypes. The statistics behind Clasnip were explained. I think there would be benefits to include a visual aid of the steps included within the Clasnip pipelines. More information should be included in the Github website for usage and setup of the pipelines.

Validity of the findings

An issue with the paper is that it doesn’t mention what other existing tools for subspecies classification exist. It says there are no standardized tools, but this implies there have been other tools (Lines 307-317). If there are, how does Clasnip compare to them with the benchmarking dataset used in this study? How is the approach used in Clasnip with HMMs and Maximum Likelihood Estimation different to these non-standard yet existing tools? What statistical models do they use?

Additional comments

In terms of wording, some sentences are a bit confusing. For example, in lines 299-301: “The input of Clasnip is not limited to short-sequences from sanger sequencing, but also the assembled long sequences from next-generation sequencing without worrying pre-alignment against reference sequences”. The latter part of this sentence is confusing. What is meant by “without worrying pre-alignment against reference sequences”? I think this part should be reworded. And maybe the sentence can be more concise? Lines 34-36 sounds like a run-on sentence and would benefit from being split into two sentences. Line 117 should have “is write” changed to “is written.” In lines 319-321, the wording is confusing, and the grammar is wrong in “fills the gap of automatically classification of subspecies”?

---

## Round 0.2 · Minor Revisions

Reviewer 2 made pertinent comments regarding Clasnip installation, in addition to minor comments. I think it's ok to require admin privileges (i.e. sudo power) to install this kind of software, but this requirement should be made clear to all potential users. I suggest you ask someone you know who has not contributed to this work to try to install the software, to ensure that remaining kinks are weeded out.

Reviewer 2 ·

Basic reporting

I thought the manuscript was easy to follow and the grammar was good. Odd wording and grammar mistakes that can be fixed quickly were rare.
DOI was added and Github repository was provided to share the raw data.

Experimental design

I think the experimental design was logical and the research question was logical. I am glad they mentioned how their tool compares to similar existing tools now.

Validity of the findings

The clasnip website seems to work fine. I am glad that documentation was added to the Github repository. I would like to let them know that I tried setting up Clasnip but I had issues working around not having administrator credentials while installing nginx (a dependency of your tool).
The documentation for nginx was very reliant on sudo, which I am not allowed to use, and attempts to use "./configure" with parameters specifiying the .conf file location and then "make" were not successful in nginx installation. It would be helpful if they include a little section for nginx because just adding the path to the .conf files under html section didn't work.

Additional comments

I include the quick minor edits here:
Abstract
Line 47 change “include” to “included”
Comparison to Related Work
Line 296. By “main” classifier you mean Clasnip, right? The sentence is worded a bit odd. You’re saying that the comparable features of Clasnip can be benchmarked with other existing sequence classification tools, right?
Lines 298-300.
Odd wording, I think “is proved on 16S rRNA gene classification with improved species-level accuracy” is a bit confusing. By proved do you mean “based on”? Or that it has gone through some sort of validation/benchmarking process on 16S rRNA gene classification?

---

## Round 0.3 · accepted · Accept

I confirm that this revised version addresses the reviewer's comments, by my own assessment. Therefore the manuscript is ready for publication.